# Fear control and danger control amid COVID-19 dental crisis: Application of the Extended Parallel Process Model

**Samane Shirahmadi[1], Shabnam Seyedzadeh-Sabounchi[1], Salman Khazaei[2], Saeid Bashirian [3]\*, Amir Farhang Miresmæili[4], Zeinab Bayat[5], Behzad Houshmand[6], Hasan Semyari[7], Majid Barati[3], Ensiyeh Jenabi[8], Fakhreddin Heidarian[9], Sepideh Zareian[9], Mohammad Kheirandish[10], Neda Dadae[10]**

1 Department of Community Oral Health, School of Dentistry and Dental Research Centers, Hamadan University of Medical Sciences, Hamadan, Iran, 2 Research Center for Health Sciences, Hamadan University of Medical Sciences, Hamadan, Iran, 3 Social Determinants of Health Research Center, Hamadan University of Medical Sciences, Hamadan, Iran, 4 Department of Orthodontics, Dental Research Center, Hamadan University of Medical Sciences, Hamadan, Iran, 5 Department of Oral Medicine, Hamadan University of Medical Sciences, Hamedan, Iran, 6 Department of Periodontics, School of Dentistry, Shahid Beheshti University of Medical Sciences, Tehran, Iran, 7 Department of Periodontics, School of Dentistry, Shahed University, Tehran, Iran, 8 Autism Spectrum Disorders Research Center, Hamadan University of Medical Sciences, Hamadan, Iran, 9 Hamadan University of Medical Sciences, Hamadan, Iran, 10 Department of Oral Health, Vice Chancellor for Health, Hamadan University of Medical Sciences, Hamadan, Iran

\* bashirian@umsha.ac.ir

**Data Availability Statement:** All relevant data are within the manuscript.

## Abstract

### Objectives

There is high risk of contamination with COVID-19 virus during routine dental procedures and infection control is crucial. The aim of this study was to determine the factors associated with Covid-19 preventive behaviors among oral health care providers using an extended parallel process model (EPPM).

### Methods

In a cross-sectional study, short text message invite surveys were sent to 870 oral health care providers in west part of Iran. Data were collected through validated self-report EPPM questionnaires. Descriptive statistics, Chi-square and Fishers exact tests were used for data analysis.

### Results

In total, 300 completed questionnaires were received and the mean age of respondents was 29.89 ± 11.17 years (range: 20–75 years). Among the study population, 284 (94.67%) perceived the threat of infection highly. Washing hands frequently with water and soap and use of hand sanitizer was reported by 93.33%, of participants. Age (P = 0.010), sex (P = 0.002) and occupation field (P = 0.010) were significantly associated with danger control responses. Data identified that those oral health care providers that were on the danger

**Funding:** The authors who received an award:
Saeid Bashirian Grant numbers awarded to author:
IR.UMSHA.REC.1398.1094. The full name of
funder: the Vice Chancellor For research and
technology from the Hamadan University of
Medical Sciences. The funders had no role in study
design, data collection and analysis, decision to
publish, or preparation of the manuscript.

**Competing interests:** The authors have declared
that no competing interests exist.

control response adopted preventive behaviors more strictly than those on fear control
response.

## Conclusion

The results of this study showed how degrees of perceived threat and perceived efficacy
influenced oral health providers' willingness to perform recommended health behaviors.
These findings can assist public health agencies in developing educational programs specif-
ically designed for promoting preventive behaviors among oral health providers in pandemic
situations.

## Introduction

Currently, the Coronavirus 2019 pandemic has become a major public health challenge [1]. As
of 20[th] of July, COVID19 has been diagnosed in 213 countries with 14,855,107 laboratory-con-
firmed cases and 613,248 deaths [2]. Current observations suggest people of all ages are usually
susceptible to this new viral disease. However, people who are in close contact with COVID-19
symptomatic and asymptomatic patients, including health care workers, are at higher risk for
SARS-CoV-2 infection [3]. Because healthcare workers are in direct contact with patients, they
may be at risk when examining and treating patients. By not following the hygienic recom-
mendations, healthcare professionals will become unwanted carriers and transmit the disease
to other patients, their family members and to the community. Currently, one in five health-
care workers have been infected with COVID-19 [4].

Meanwhile, dentistry is one of the riskiest occupations. Since there is a high possibility of
spread of and exposure to blood, saliva, body fluids, and respiratory secretions while providing
dental services. Due to the characteristics of the dental settings, mutual infection between den-
tists and patients are possible. Therefore, the principle of infection control should be strictly
enforced, to provide safety for both dentists and patients [5, 6]. Normal protective measures in
daily clinical practice are not effective enough to prevent the spread of COVID-19 since a large
quantity of droplets and suspended particles are created while removing decay or scaling and
cleaning [3]. Therefore, it is important to develop and follow accurate and effective preventive
protocols. Dentists need to be encouraged to follow personal protection and infection control
protocols carefully and thoroughly. The results of a study by Ahmed et al. on dental practice
during the COVID-19 pandemic showed that 90% of the participating dentists were aware of
changes in the treatment protocols. Nonetheless, only 61% of them performed the improved
treatment protocols correctly [7].

Research has shown that risk messages distributed through health care providers can cause
fear in patients and lead to positive changes in their attitude, intention and protective behavior
responses [8]. However, one key factor in designing such educational programs is maximizing
efficacy and avoiding unintended consequences such as fear and anxiety and disease-related
stigma [9]. As the results of the study of Ahmad et al., despite having a high level of knowledge
and practice, dentistry around the world suffers from anxiety and fear when working in the
relevant fields due to the impact of the COVID-19 epidemic on humanity. This anxiety and
fear has led some dentists to change their dental procedures or services in accordance with the
recommended guidelines for emergency treatment or to close their clinics for an indefinite
period [7].

Extended Parallel Processing Model (EPPM) is one of the models that predicts the performance of health behaviors recommended by a person by analyzing fear appeals [8]. According to Extended Parallel Processing Model (EPPM), when people confront a serious threat, they are afraid and choose one of the two danger control or fear control responses to control the threat. When a person's motivation is danger control, he/she consciously thinks about the danger and ways to prevent the threat (protection motivation). In contrast, fear control processes focus on controlling internal concerns (such as emotions and physiological responses) [8].

The two main elements of danger perception in this model are perceived threat and perceived efficacy (Fig 1). Perceived threat consists of two basic dimensions: severity (susceptibility to the importance or magnitude of the threat) and susceptibility (the person's belief in vulnerability to the threat). Perceived efficacy has two basic dimensions: response efficacy (effectiveness of the recommendations provided in preventing or counteracting the perceived threat) and self-efficacy (a person's belief in his or her ability to follow advice) [8].

EPPM suggests that there is relationship between threat and multiplicative efficacy. So that in the conditions of high threat/high efficacy, the processes and consequences of danger control develop positive changes in attitudes, goals and behaviors. In high threat/low efficacy

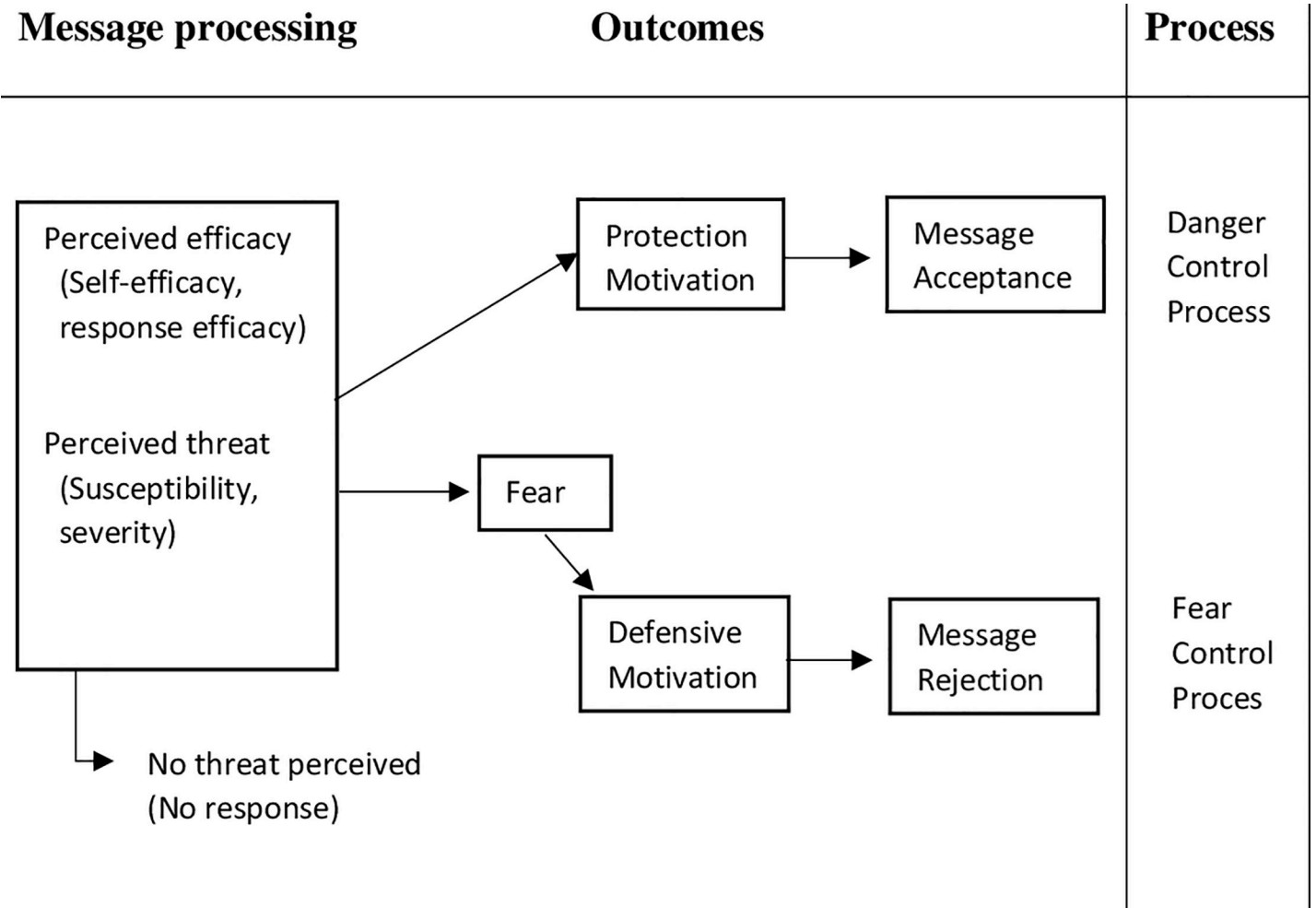

**Fig 1. The Extended Parallel Process Model (EPPM) [8].**

conditions, fear control processes and their outcomes are initiated including defensive avoidance or reactance.

Due to the outbreak of coronavirus infection in Iran and the high importance of infection control in dentistry, the aim of this study was to determine the factors associated with Covid-19 preventive behaviors in oral care health providers' using a model of EPPM. The results of this study can be used to design and implement appropriate educational interventions in order to prevent and control COVID-19 disease in dental settings. Our objectives were to answer the following questions and null hypotheses:

- What is the percentage of oral health care providers who are on fear control and danger control response?

- What is the influence of different demographic characteristics on health behaviors among oral health care providers regarding COVID-19 epidemic?

- There are no associations between oral health care providers' demographic characteristics and perceived threat and efficacy to COVID-19 epidemic.

- There is no association between oral health behavior and perceived threat and efficacy to COVID-19 epidemic among oral health care providers.

- There are no associations between oral health care providers' demographic characteristics and danger control responses to COVID-19 epidemic.

## Materials and methods

### Design

This cross-sectional study was conducted between February 24 and March 30 in the year 2020, in the west part of Iran. In this study, factors affecting COVID-19 preventive behaviors in dentists was investigated using an Extended Parallel Processing Model. This survey was conducted in Farsi

### Sample and setting

In fifth of March 2020, a text message invitation to take a survey was sent to 860 dentists, faculty, and graduate or postgraduate dental students, dental assistants, and dental hygienists, in the city of Hamadan. This invitation message was sent via Short Message Service (SMS) on national registered phone users and WhatsApp application to a list of all registered dentists in the general dental council. For those who did not complete the questionnaire within 10 days, the invitation message was resent again on March 15[th], 2020. Data were collected for 10 days until March 30[th], 2020. The software application was designed in such a way that the oral health care providers needed to answer all questions, otherwise, they did not receive a final confirmation.

### The survey instrument

The data collection instrument included a questionnaire consisting of two general sections of demographics and EPPM structures, which were completed by the participants in the study. We designed and constructed the EPPM scale based on questionnaires used by previous peer-reviewed studies [10–12]. In these studies, the validity and reliability of the questionnaires were measured through Exploratory Factor Analysis (EFA), Confirmatory Factor Analysis (CFA), Cronbach's alpha coefficient and the Intra class Correlation Coefficient (ICC) [10–12].

The validity measures and Cronbach's alpha coefficient of the designed questionnaire in this study are similar to previous studies. However, the Cronbach's alpha constructs in this study were eventually re-examined and improved [10–12]. Eight health education, and promotion specialists and 2 dentists took part in the validation procedure. The CVR and CVI scores were 0.80 and 0.79, respectively. Face validity of the questionnaire was assessed by 30 dentists whose characteristics were similar to the target study sample.

The reliability of the questionnaire was assessed by calculating the internal reliability. Alpha Cronbach was 0.70 for "perceived susceptibility", 0.78 for "perceived severity", 0.84 for "self-efficacy", 0.87 for "response efficacy" and 0.92 for "preventive behaviors."

Demographic variables included in the questionnaire were age, gender, previous work experience (year), occupation field (dentist, graduate dent student, postgraduate dental student, dental hygienist). The EPPM part of the questionnaire was consisted of 30 questions which covered five constructs as described in Table 1.

## Data analysis

The statistical analysis was performed using SPSS version 16.0 software. Using descriptive statistics (mean, standard deviation, frequency and percentage) in the form of tables and graphs, the study population was described. The Chi-Square test and Fisher Accurate tests were used to compare the level of threat and perceived efficacy based on demographic variables.

Using Likert scale responses, the "threat" construct was developed as a product of participants' response to perceived susceptibility and perceived severity. The "efficacy" construct was identified through calculating the participants' responses to self-efficacy and efficacy. Witte discriminating value was used to measure danger control responses [13] and with Chi-square test, the comparison of demographic variables was performed at the levels formed by the model (danger control and fear control response).

Low and high values of threat and perceived efficacy were determined by the median value of each structure, respectively. The four EPPM classifications were created based on the perceived threat level and the perceived efficacy level. These classes included low threat and efficacy (LT/LE), low threat and high efficacy (LT/HE), high threat and low efficacy (HT/LE), and finally high threat and efficacy (HT/HE). Using the Fisher Accurate Test, the performance of the seven preventive behaviors examined at the levels formed by the model (LT/LE, HT/LE, LT/HE, HT/HE) were compared. Furthermore, 7 preventive behaviors were compared by using Chi-square test according to the levels formed by the model (danger control and fear control paths). We reduced the preventive behavior responses to always and sometimes since the number of never responses were low. Significance level in all tests was considered at p<0.05.

**Table 1. EPPM questionnaire about health behaviors regarding COVID-19 epidemic in oral health care providers.**

| Construct | Questions (N) | Examples | Scale |
|---|---|---|---|
| **Perceived Severity** | 3 | Coronavirus can cause death | 5-point Likert Scale from 1 (completely disagree) to 5 (completely agree) |
| **Perceived Susceptibility** | 2 | I am unlikely to get coronavirus | 5-point Likert Scale from 1 (completely disagree) to 5 (completely agree) |
| **Response Efficacy** | 11 | Wearing goggles, masks or shields during all stages of treatment is effective in preventing coronavirus. | 5-point Likert Scale from 1 (completely disagree) to 5 (completely agree) |
| **Self-Efficacy** | 7 | I can use goggles, masks or shields during all stages of treatment | 5-point Likert Scale from 1 (completely disagree) to 5 (completely agree) |
| **Preventive Behavior** | 7 | I wear goggles and shield masks during all stages of treatment | 1–3 Likert scale which 1 represented never and 3 was always |

### Ethical consideration

The Ethics Committee of Hamadan University of Medical Sciences approved this study (IR). UMSHA.REC.1398.1094). All data collected from the study participants were de-identified. No direct benefits or rewards were paid to participants for their participation in this study.

### Results

In total, 300 (34.8%) completed electronic questionnaires were received. The mean age of the participants in this study were 29.89 ± 11.17 years (range: 20–75 years). Table 2 shows the demographic characteristics of the participants in this study. Two hundred and eighty four (94.67%) and 285 (95%) of the study populations had high perceived threat and efficacy, respectively (Table 2). The proportion of high efficacy was significantly higher is employees with more than 5-years of previous work experience (P = 0.040, Table 2).

Among the participants 93.3% washed their hands with water and soap frequently or used hand disinfectant (Fig 2).

In accordance with the EPPM model, the proportion of participations with low perceived threat and efficacy (LT/LE), low threat-high efficacy (LT/HE), High threat-low efficacy (HT/LE) and both high perceived threat and efficacy (HT/HE) were 0.67%, 4.67%, 4.33% and 90.33%, respectively (Table 3). Oral health care providers with high threat-high efficacy significantly performed behavior 6 (disinfect equipment and surfaces regularly) more than other categories (P = 0.048, Table 3).

Age (P = 0.010), sex (P = 0.002), and field of occupation (P = 0.011) were significantly associated with the danger control response (Fig 3).

### Discussion

The results of the present study identified that the perceived threat and efficacy construct among the study participants were at a high level (94.6% high perceived threat and 95% high perceived efficacy). Females, younger participants, those with less previous work experience and dental hygienists were more motivated to control their fears than other groups.

The results of this study showed that previous work experience has a significant relationship with the perceived efficacy. Participants with higher previous work experience were more efficient. Similar results have been reported by Wilson [14] and Raybould [15]. The most

**Table 2. Associations between respondents' demographic characteristics and perceived threat and efficacy to COVID-19 epidemic.**

| | Variable | Total N (%) | | Threat | P value | | Efficacy | P value |
|---|---|---|---|---|---|---|---|---|
| | | | Low N (%) | High N (%) | | Low N (%) | High N (%) | |
| Gender | Male | 160 (53.3) | 10(6.2) | 150 (93.7) | 0.451 | 6(3.7) | 154(96.2) | 0.292 |
| | Female | 140(46.6) | 6(4.2) | 134 (95.7) | | 9(6.4) | 131(93.5) | |
| Age group (year) | 20–29 | 211 (70.3) | 12(5.6) | 199(94.3) | 0.892 | 15(7.1) | 196(92.8) | 0.080 |
| | 30–39 | 37(12.3) | 1(2.7) | 36(97.3) | | 0 | 37(100.0) | |
| | 40–49 | 14(4.6) | 1(7.1) | 13(92.8) | | 0 | 14(100.0) | |
| | 50–60 | 38(12.6) | 2(5.2) | 36(94.7) | | 0 | 38(100.0) | |
| Previous work experience (year) | <5 | 212(70.6) | 12(5.6) | 200(94.3) | 0.595 | 15(7.0) | 197(92.9) | **0.040** |
| | 5–10 | 44(14.6) | 1(2.2) | 43(97.7) | | 0 | 44(100.0) | |
| | >10 | 44(14.6) | 3(6.8) | 41(9.1) | | 0 | 44(100.0) | |
| Occupation Field | Dentist | 175(58.3) | 6(3.4) | 169(96.5) | 0.221 | 6(3.4) | 169(96.5) | 0.224 |
| | Student | 98(32.6) | 8(8.1) | 90(91.8) | | 8(8.1) | 90(91.8) | |
| | Dental hygienist | 27(9.0) | 2(7.4) | 25(92.5) | | 1(3.7) | 26(96.3) | |
| | Total | 300(100.0) | 16(5.4) | 284(94.6) | - | 15(5.0) | 285(95.0) | - |

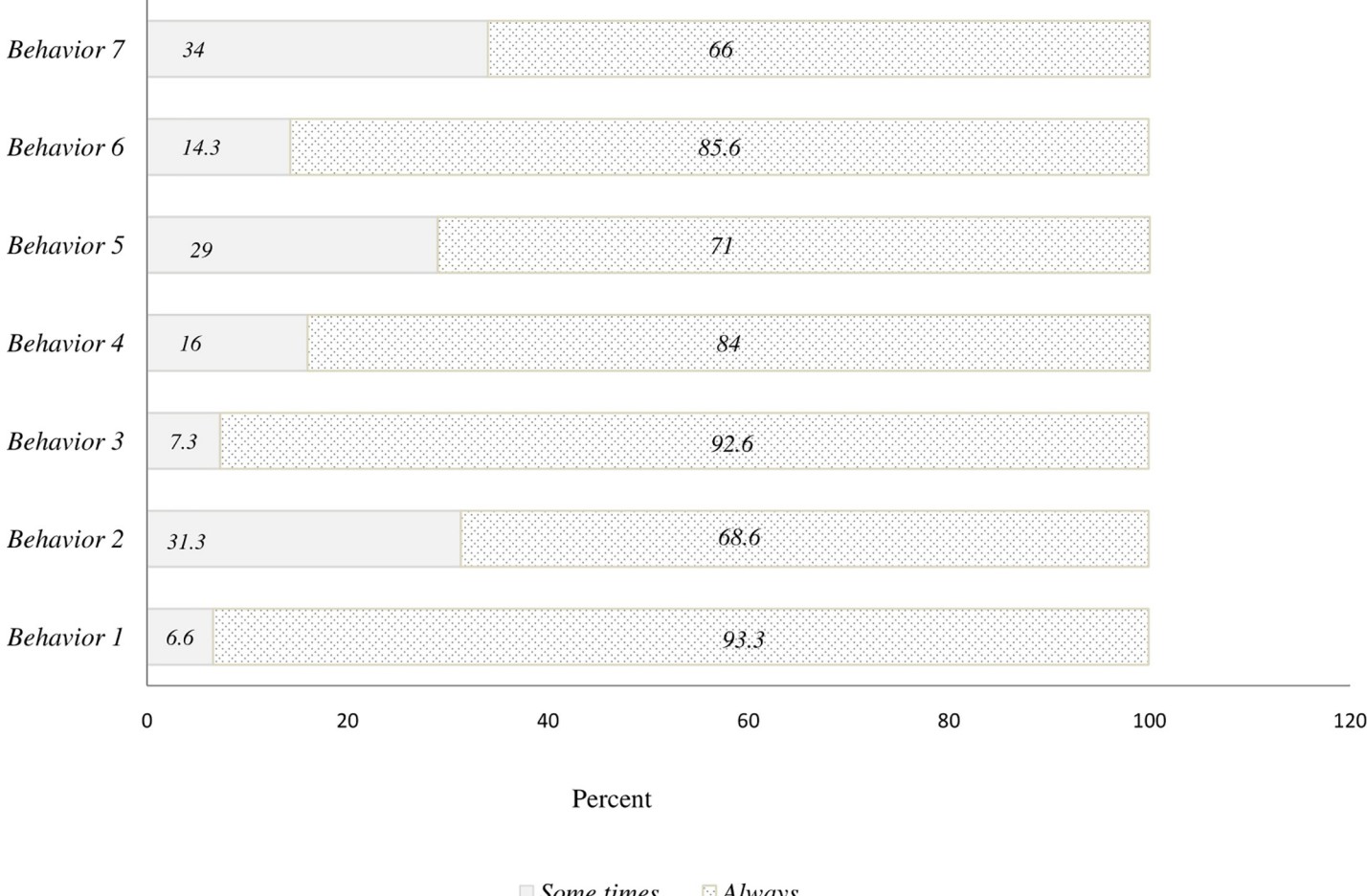

**Fig 2. Health behaviors of oral health care providers regarding prevention of COVID-19.** Behavior 1: washing hands frequently with water and soap/use hand sanitizer, behavior 2: Do not admit suspicious patients in the Dental office, behavior 3: Avoid close contact, behavior 4: use protection for eyes such as goggles, masks or shields during all treatment steps, behavior 5: use protective uniforms (gown or protective clothing, protective gloves) during all treatment steps, behavior 6: Disinfect equipment and surfaces regularly, behavior 7: Request patients to wash their hands with soap and water upon entering the dental clinic.

important influencing source on efficacy is past experience according to the literature [16]. Therefore, previous experience of dealing with various epidemics, including influenza and infectious diseases such as AIDS and hepatitis, can increase the efficacy of providers with longer previous work experience [17].

The trends observed in previous studies [18, 19] and the results of this study show that the efficacy dimension of EPPM has a more positive effect on health behavior than the threat dimension. This pattern was also observed in preventive behaviors numbers 4 to 7 (use protection for eyes such as goggles, masks or shields during all treatment steps, wear protective uniforms (gown or protective clothing, protective gloves) during all treatment steps, disinfect equipment and surfaces and regularly, request patients to wash their hands with soap and water upon entering the dental office). This indicates a more positive effect of the perceived efficacy dimension than the perceived threat dimension on the recommended health behaviors. The results of a study conducted in Israel also showed that lower psychological distress was associated with having higher scores for self-efficacy [20]. Therefore, it seems that in viral epidemics, in order to intervene and provide guidance, more emphasis should be placed on the efficacy dimension.

**Table 3. Health behaviors regarding COVID-19 epidemic in oral health care providers in different levels of the EPPM.**

| Behavior | | Low threat-Low efficacy, N (%) | Low threat-High efficacy, N (%) | High threat-Low efficacy, N (%) | High threat-High efficacy, N (%) | P value |
|---|---|---|---|---|---|---|
| Behavior 1 | Some times | 0 | 1 (7.1) | 1 (7.6) | 18 (6.6) | 0.873 |
| | Always | 2 (100.0) | 13 (92.8) | 12 (92.3) | 253 (93.3) | |
| Behavior 2 | Some times | 1 (50.0) | 7 (50.0) | 3 (23.0) | 83 (30.6) | 0.325 |
| | Always | 1 (50.0) | 7 (50.0) | 10 (76.9) | 188 (69.3) | |
| Behavior 3 | Some times | 0 | 2 (14.2) | 1 (7.6) | 19 (7.0) | 0.542 |
| | Always | 2 (100.0) | 12 (85.7) | 12 (92.3) | 252 (92.9) | |
| Behavior 4 | Some times | 0 | 3 (21.4) | 3 (23.0) | 41 (15.1) | 0.401 |
| | Always | 2 (100.0) | 11 (78.5) | 10 (76.9) | 230 (84.8) | |
| Behavior 5 | Some times | 0 | 2 (14.2) | 4 (30.7) | 81 (29.8) | 0.584 |
| | Always | 2 (100.0) | 12 (85.7) | 9 (69.2) | 190 (70.1) | |
| Behavior 6 | Some times | 0 | 3 (21.4) | 5 (38.4) | 35 (12.9) | **0.048** |
| | Always | 2 (100.0) | 11 (78.5) | 8 (61.5) | 236 (87.0) | |
| Behavior 7 | Some times | 0 | 5 (35.7) | 5 (38.4) | 92 (33.9) | 0.891 |
| | Always | 2 (100.0) | 9 (64.2) | 8 (61.5) | 179 (66.0) | |
| Total | | 2 (0.67) | 14 (4.67) | 13 (4.33) | 271 (90.33) | |

Behavior 1: washing frequently hands with water and soap and a hygienic hand disinfection, behavior 2: Do not visit the patient or suspicious people in the Dental office, behavior 3: Avoid close contact, behavior 4: use protection for eyes such as goggles, masks or shields during all treatments, behavior 5: use of protective uniforms (gown or protective clothing, protective gloves) during all treatment steps, behavior 6: Disinfect equipment and surfaces, behavior 7: Requiring patients to wash their hands with soap and water upon entering the Dental clinic.

The Centers for Disease Control and Prevention (CDC), the American Dental Association (ADA) and the World Health Organization (WHO), strongly advise oral health care providers to discontinue unnecessary dental treatments during the epidemic period and not to treat or examine people suspected of having the disease in regular dental clinics [21–23]. The ministry of health and medical education in Iran has made similar recommendations to its oral health care providers [24]. The results of this study show that although 90.3% of the participants in the study were in the HT/HE class, only 69.3% of them avoided to visit people with suspected disease in their office. Consolo et al. also reported the same results [12]. The reason for this avoidance can be that they had high perceived response efficacy since 95% of the participants considered the recommended health behaviors to be effective in preventing the transmission of the disease and perceived their ability as high to perform the preventive behaviors.

Other recommendations that the Ministry of Health and Medical Education of Iran has for dentists during this period include: strict use of personal protective equipment, hand washing, detailed patient evaluation, using only anti-retraction hand piece, asking patients rinse their mouths before dental procedures, and disinfecting surfaces in the clinic [24]. The results of this study showed that more than 70% of the participants in the study observed the recommended behaviors. Consolo et al. also reported the same results [12]. The reason could be that oral health care providers have been always required to follow the infection control guidelines even before the COVID-19 pandemic.

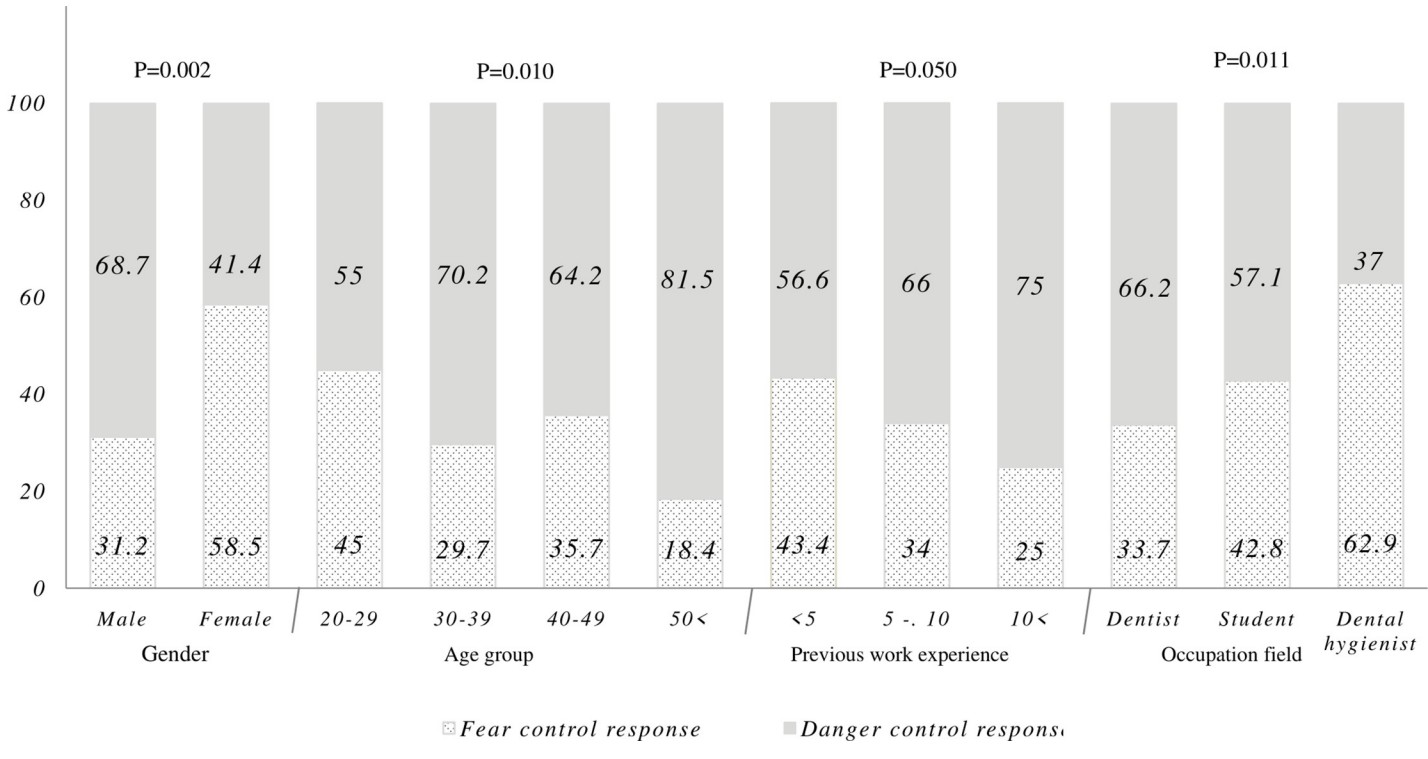

**Fig 3. Results of the discriminating value formula for danger control responses.**

As of 21st of July 2020, COVID 19 had been diagnosed in Iran with 276,202 laboratory confirmed cases and 14,405 deaths had been recorded [2]. Based on the high number of COVID-19 positive patients in Iran, and the high risk of dental setting in terms of transmission and spread of the corona virus [3], it seems there is a need for accurate monitoring of oral health care providers performance on preventing transmission and spread of the disease.

In this study, the calculated discriminating value determined whether participants were motivated to control fear or to control danger similar to previous studies [13]. The results identified a statistically significant difference between the different groups studied in terms of fear control and danger control. This shows that the members of different occupation field groups respond differently when faced with a health problem, such as COVID 19. Therefore, differences between groups should be taken into consideration while designing interventions based on groups' specific characteristics and conditions.

This study has several important strengths. Given that the health messages distributed in this COVID-19 pandemic situation has not successfully influenced perceptions, this study determined the control processes. Based on the results, specific target groups can be identified and the necessary interventions can be tailored according to the control processes.

According to the results of this study, Females, younger participants and dental hygienists were more motivated to control their fears than other groups. Interventions can be designed to increase their perceived efficacy. These interventions include educating providers on coronavirus disease-related guidelines and recommendations for conducting health behaviors using health education models. The focus of the models can be directed towards increasing self-efficacy, such as self-efficacy model, self-regulation model and using implementation intention theory.

Also, environmental interventions can be implemented such as providing personal protective equipment (PPE) for students and dental hygienists, and strict adherence to infection control protocols in schools of dentistry and dental clinics. In dental schools and clinics careful monitoring of the implementation of infection control protocols and the use of PPE, establishment of Safety and Hygiene Scheme 2020 (SHS 2020) are highly recommended through health education planning models such as PRECEDE-PROCEED Model and intervention mapping.

One of the limitations of this study was that questionnaires were self-administered and could induce social desirability. Since participants may not report the actual behavior but the behavior that would be desired by the society. The low number of items in some constructs was another limitation of this study. Also, the findings of our study on oral health care providers may not necessarily lead to similar findings among other health care providers such as hospital staff and nurses.

Cross-sectional studies do not determine the causal relationship between the recommended health behaviors and the levels formed by the model (danger control and fear control). However, cross-sectional studies are an important tool for identifying danger and protection indicators for future longitudinal assessments.

## Conclusion

The results of this study showed how degrees of perceived threat and perceived efficacy affect oral health care providers' willingness to perform recommended health behaviors. The findings showed that when perceived efficacy of the recommended health behaviors overcame the perceived threat, the likelihood of preventive health behaviors regarding COVID-19 increased. Therefore, a theory-based behavioral modification program can be developed based on gender among dental students and dental hygienists. Older oral health care providers and those with more years of experience require intense educational interventions to modify their hygienic behaviors compared to younger providers.

## Acknowledgments

The research team would like to thank the dentists, students and dental hygienists who have participated in this study.

## Author Contributions

**Conceptualization:** Saeid Bashirian.

**Data curation:** Samane Shirahmadi, Salman Khazaei, Saeid Bashirian.

**Formal analysis:** Samane Shirahmadi, Salman Khazaei, Saeid Bashirian.

**Funding acquisition:** Saeid Bashirian.

**Investigation:** Samane Shirahmadi, Shabnam Seyedzadeh-Sabounchi, Amir Farhang Miresmæili, Zeinab Bayat, Behzad Houshmand, Hasan Semyari, Majid Barati, Ensiyeh Jenabi, Fakhreddin Heidarian.

**Methodology:** Samane Shirahmadi, Saeid Bashirian.

**Project administration:** Saeid Bashirian.

**Resources:** Samane Shirahmadi, Fakhreddin Heidarian, Mohammad Kheirandish, Neda Dadae.

**Software:** Samane Shirahmadi, Salman Khazaei, Sepideh Zareian.

**Supervision:** Samane Shirahmadi, Saeid Bashirian.

**Validation:** Samane Shirahmadi, Shabnam Seyedzadeh-Sabounchi, Amir Farhang Miresmæili, Zeinab Bayat, Behzad Houshmand, Hasan Semyari, Majid Barati, Fakhreddin Heidarian.

**Writing – original draft:** Samane Shirahmadi, Shabnam Seyedzadeh-Sabounchi, Salman Khazaei, Saeid Bashirian.

**Writing – review & editing:** Samane Shirahmadi, Shabnam Seyedzadeh-Sabounchi, Salman Khazaei, Saeid Bashirian, Amir Farhang Miresmæili, Zeinab Bayat, Behzad Houshmand, Hasan Semyari, Majid Barati, Ensiyeh Jenabi, Fakhreddin Heidarian, Sepideh Zareian, Mohammad Kheirandish, Neda Dadae.

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
