## [Decision Letter · Decision Letter 0]

22 Jun 2020

PONE-D-20-14093

Fear Control and Danger Control Amid COVID-19 Dental Crisis: Application of the Extended Parallel Process Model

PLOS ONE

Dear Dr. Bashirian,

Thank you for submitting your manuscript to PLOS ONE. After careful consideration, we feel that it has merit but does not fully meet PLOS ONE’s publication criteria as it currently stands. Therefore, we invite you to submit a revised version of the manuscript that addresses the points raised during the review process.

With your revised draft, please pay special attention to all reviewers' comments, and revise your paper in accordance with PLOS ONE's Guidelines for Authors. Please remember that an insufficiently revised draft must lead to outright reject.

We look forward to receiving your revised manuscript.

Kind regards,

Andrej M Kielbassa, Prof. Dr. med. dent. Dr. h. c.

Academic Editor

PLOS ONE

Journal Requirements:

2. Please include additional information regarding the survey or questionnaire used in the study and ensure that you have provided sufficient details that others could replicate the analyses.

For instance, if you developed a questionnaire as part of this study and it is not under a copyright more restrictive than CC-BY, please include a copy, in both the original language and English, as Supporting Information.

3.PLOS requires an ORCID iD for the corresponding author in Editorial Manager on papers submitted after December 6th, 2016. Please ensure that you have an ORCID iD and that it is validated in Editorial Manager. To do this, go to ‘Update my Information’ (in the upper left-hand corner of the main menu), and click on the Fetch/Validate link next to the ORCID field. This will take you to the ORCID site and allow you to create a new iD or authenticate a pre-existing iD in Editorial Manager. Please see the following video for instructions on linking an ORCID iD to your Editorial Manager account: https://www.youtube.com/watch?v=_xcclfuvtxQ

Reviewers' comments:

Reviewer's Responses to Questions

**Comments to the Author**

1. Is the manuscript technically sound, and do the data support the conclusions?

Reviewer #1: Yes

Reviewer #2: Partly

Reviewer #3: No

2. Has the statistical analysis been performed appropriately and rigorously? 

Reviewer #1: Yes

Reviewer #2: No

Reviewer #3: No

3. Have the authors made all data underlying the findings in their manuscript fully available?

Reviewer #1: Yes

Reviewer #2: Yes

Reviewer #3: No

4. Is the manuscript presented in an intelligible fashion and written in standard English?

Reviewer #1: Yes

Reviewer #2: No

Reviewer #3: No

5. Review Comments to the Author

Reviewer #1: Dear Authors

See the attached comments on the PDF file. These all are for the improvement and betterment of this manuscript with latest information.

Extract information from the below papers in the manuscript;

a) Peng, Xian, et al. "Transmission routes of 2019-nCoV and controls in dental practice." International Journal of Oral Science 12.1 (2020): 1-6.

b) Ahmed, M.A.; Jouhar, R.; Ahmed, N.; Adnan, S.; Aftab, M.; Zafar, M.S.; Khurshid, Z. Fear and Practice Modifications among Dentists to Combat Novel Coronavirus Disease (COVID-19) Outbreak. Int. J. Environ. Res. Public Health 2020, 17, 2821.

c) Khurshid, Zohaib, Faris Yahya Ibrahim Asiri, and Hamed Al Wadaani. "Human saliva: non-invasive fluid for detecting novel Coronavirus (2019-nCoV)." International Journal of Environmental Research and Public Health 17.7 (2020): 2225.

Also, improve and double check your references.

Reviewer #2: I have been pleased to review this interesting paper, "Fear Control and Danger Control Amid COVID-19 Dental Crisis: Application of the Extended Parallel Process Model". The topic is important nowadays and the authors provide an interesting aspect to it, through usage of the extended parallel process model.

However, I have several comments that need to be answered.

1. The compliance rate of this cross-sectional survey seems to be approx. 34%. Can the authors explain such compliance rate?

2. English editing and proofing of the manuscript should be performed, e.g. in the discussion "Participants with higher previous work experiencewere (experience were) more efficient" etc.

3. The authors refer to "The centers for disease control and prevention (CDC) the national association of dentists strongly advise dentists to discontinue unnecessary dental treatments during the epidemic period and not to treat or examine people suspected of having the disease in regular dental clinics (15, 16). However, there is no information regarding the instructions given in Iran. What were the instructions provided by the local dental board? Were dentists instructed to continue with elective dental care or provide emergency treatments only? It could be useful to add such information to the discussion.

4. What were the exact dates of survey conductance in March 2020?

5. What was the scope (e.g. number of COVID-19 cases and deaths per capita) of the COVID-19 pandemic in Iran during the survey date of conductance? It could be useful to add such information to the discussion.

6. Up to date, several studies regarding dentists' psychological distress and self-efficacy in the COVID-19 era. The authors should relate to such up-to-date studies (e.g.

Shacham, M., Hamama-Raz, Y., Kolerman, R., Mijiritsky, O., Ben-Ezra, M., & Mijiritsky, E. (2020). COVID-19 factors and psychological factors associated with elevated psychological distress among dentists and dental hygienists in Israel. International Journal of Environmental Research and Public Health, 17(8), 2900.

Consolo, U., Bellini, P., Bencivenni, D., Iani, C., & Checchi, V. (2020). Epidemiological Aspects and Psychological Reactions to COVID-19 of Dental Practitioners in the Northern Italy Districts of Modena and Reggio Emilia. International Journal of Environmental Research and Public Health, 17(10), 3459.)

7. "a text message invitation to take a survey was sent". How was it sent? WhatsApp? Were participants contacted more than once?

8. Some parts of the questionnaire has very low number of items, e.g. perceived sustainability, along with internal reliability values rated as good. These should be added to the limitations of the study.

9. In what language was the survey conducted? English, Arabic, Farsi, other?

10. Were the questions adopted from previous validated questionnaires?

11. "Significance level in all tests was considered at p<0.05". Perhaps the authors meant consideration at p≤0.05? Since in the results several parameters were

12. "The proportion of high efficacy was significantly higher is employees with more than 5-years of previous work experience (P=0.04) and those older than 30 years of age (P=0.08)." "more than other categories (P=0.055)." How come P=0.08 and P=0.55 are considered significant, if values were considered significant <0.05?

Reviewer #3: Abstract

- English remains a concern. Please revise for language shortcomings, to facilitate reading. See, for example, "procedres".

- "(...) and high efficacy towards coronaviruse disease, respectively." Meaning is unclear, please revise.

- "The proportion of high efficacy was significantly higher (...)." Meaning remains unclear.

- "Data identified that those oral health care providers that were on the risk control response (...)." Again, what is risk control response?

- "(...) adopted preventive behaviors more strictly r than those on fear control response." Check sentence, and clarify the term "fear control response".

Intro

- Meaning ov "observaning"?

- Again, please revise for a clear language. Provide complete sentences. Note that the current draft must be thoroughly revised, and help of a native speaker is strongly encouraged. With the numerous (co-)authors of the current submission, quality of the submitted manuscript clearly would seem questionable. Remember that ALL authors must have read (and should have revised) this submitted draft.

- "In infectious disease 89 epidemic campaigns, main focus is on motivating fear." Meaning would seem unclear. Who motivates fear? Provide references.

- Authors have missed to elaborated both aims and objectives. This section is not considered satisfying.

- A clear and indisputable null hypothesis would seem missing.

Meths

- Numerous aspects do not belong to the Methodology section here. Remember not to provide a literature review with this section. Please revise thoroughly, and provide references. Your explanation of EPPM should be transferred to the Intro section.

- "In total, 300 (34.8%) completed electronic questionnaires were received." This would seem to present a results, right?

- Please compare: "0.78 for "perceived severity" 0.78, 0.84 for 140 "self-efficacy", (...)", and you will understand that it seem doubtful whether all co-authors have read and approved this manuscript.

- "was consisted of 30 questions which covered 144 five constrcuts": Again, please revise thoroughly, and eliminate all language shortcomings.

- What is "susestbility"?

Results

- Again, this section has not been convincingly elaborated. Numerous typos should be eliminated, to facilitate reading.

Disc

- As with the other sections, proper elaborations would seem mandatory. See "efficacy" vs "efficiency".

- "COVID 19 infection had been selected since it is currently an international public health problem." This is not considered a "strength" of the submitted study/paper.

- "Based on the results, specific target groups can be identified and the necessary interventions can be tailored according to the control processes." Please provide these informations. What "specific target groups", and "what necessary interventions to be tailored" do you refer to?

- What is "bias induced by social utility"? Please explain.

Concl

- Please do not simply repeat your results here. Instead, provide a reasonable extension of your outcome.

- Stick exclusively to your aims, and focus on your outcome. Do not provide general phrases or common places here.

Tables & Figs

- Do not repeat or double your results.

In total, this draft would seem in need of a thorough revision, and is not considered ready to proceed.

6. PLOS authors have the option to publish the peer review history of their article (what does this mean?). If published, this will include your full peer review and any attached files.

Reviewer #1: No

Reviewer #2: No

Reviewer #3: No

---

## [Author Response · Author response to Decision Letter 0]

14 Jul 2020

Author’s response to reviews

Title: Fear Control and Danger Control Amid COVID-19 Dental Crisis: Application of the Extended Parallel Process Model

Authors:

Samaneh Shirahmadi (shirahmadi_s@yahoo.com)

Shabnam Seyedzadeh-Sabounchi, 

Salman Khazaei, 

Saeid Bashirian, 

Amir Farhang Miresmæili, 

Zeinab Bayat, 

Behzad Houshmand

Hasan Semyari

Majid Barati, 

Ensiyeh Jenabi, 

Fakhreddin Heidarian, 

Sepideh Zareian, 

Mohammad Kheirandish, 

Neda Dadae

Version: 1 Date: 11 July 2020

Author's response to reviews: see over

We thank all the Reviewers for their valuable feedback and taking the time to provide useful comments to improve our manuscript entitled “Fear Control and Danger Control Amid COVID-19 Dental Crisis: Application of the Extended Parallel Process Model”. 

Based on the constructive comments the following changes have been made. 

Response to Reviewer 1:

1. Here authors have to add about saliva as source of transmission during dental practice. 

Response: Thank you for your valuable comment. We have added saliva as source of transmission during dental practice (Rreference 6). Page3, line 85.

2. Write the outcome of this survey here to strengthen the manuscript. 

Response: Thank you for your valuable comment. We have added these sentences in the introduction:

"The results of a study by Ahmed et al. on dental practice during the COVID-19 pandemic showed that 90% of the participating dentists were aware of changes in the treatment protocols. Nonetheless, only 61% of them performed the improved treatment protocols correctly." Page 4, lines 90-97. 

Response to Reviewer 2:

1. The compliance rate of this cross-sectional survey seems to be approx. 34%. Can the authors explain such compliance rate?

Response: Thank you for your valuable comment. Currently multiple studies are being conducted simultaneously by different healthcare sectors and researchers on the impact of the COVID-19 pandemic. It seems that the multiplicity of similar questionnaires distributed could be a reason for the low response rate (RR). Infection control in dentistry has been a requirement for dentists even before the COVID-19 pandemic. Another reason for low RR might be that dentists did not perceive the importance of this survey as critical. In addition, based on literature low response rate among health care providers is also common (1).

1. Wiebe ER, Kaczorowski J, MacKay J. Why are response rates in clinician surveys declining? Canadian Family Physician. 2012;58(4):e225-e8.

2. English editing and proofing of the manuscript should be performed, e.g. in the discussion "Participants with higher previous work experience were (experience were) more efficient" etc.

Response: We have improved English language.

3. The authors refer to "The centers for disease control and prevention (CDC) the national association of dentists strongly advise dentists to discontinue unnecessary dental treatments during the epidemic period and not to treat or examine people suspected of having the disease in regular dental clinics (15, 16). However, there is no information regarding the instructions given in Iran. What were the instructions provided by the local dental board? Were dentists instructed to continue with elective dental care or provide emergency treatments only? It could be useful to add such information to the discussion.

Response: Thank you for your valuable comment. We have added these sentences in discussion:

"While the ministry of health and medical education in Iran has made similar recommendations to its dentists". Page14, Line328.

And 

Other recommendations that the Ministry of Health and Medical Education of Iran has for dentists during this period include: strict use of personal protective equipment, hand washing, detailed patient evaluation, rubber dam isolation, using only anti-retraction handpiece, asking patients rinse their mouths before dental procedures, and disinfecting surfaces in the clinic (1). Based on the high number of COVID-19 positive patients in Iran (As of 25th of March 2020, COVID 19 had been diagnosed in Iran with 27017 laboratory-confirmed cases and 2077 deaths had been recorded (2)), and the high risk of dental setting in terms of transmission and spread of the corona virus (3), it seems there is a need for accurate monitoring of oral health care providers performance on preventing transmission and spread of the disease. Page15, Line336-348.

1. Iran Tmohamei. Protocol for the provision of dental services and related letters in the context of the COVID-19 epidemic. Tehran2020.

2. Organization wh. Coronavirus disease 2019 (COVID-19): Situation Report - 65 (25 March 2020) 2020. Available from: https://reliefweb.int/report/world/coronavirus-disease-2019-covid-19-situation-report-65-25-march-2020.

3. Meng L, Hua F, Bian Z. Coronavirus Disease 2019 (COVID-19): Emerging and Future Challenges for Dental and Oral Medicine. Journal of Dental Research. 2020;99(5):481-7.

4. What were the exact dates of survey conductance in March 2020?

Response: Thank you for your valuable comment. We have improved the method section and provided the exact dates of survey:" This cross-sectional study was conducted between February 24 and March 30 in the year 2020, in the west part of Iran." Page6, Line153. Also, we have added this sentence in the Sample and Setting section:"Data were collected for 10 days until March 30, 2020. Page7,177-183. Line…..

5. What was the scope (e.g. number of COVID-19 cases and deaths per capita) of the COVID-19 pandemic in Iran during the survey date of conductance? It could be useful to add such information to the discussion.

Response: Thank you for your valuable comment. We have added this sentence "As of March 25th 2020, in Iran 27,017 laboratory-confirmed COVID 19 positive cases and 2,077 deaths had been recorded" in discussion section. Page15, Line344.

6. Up to date, several studies regarding dentists' psychological distress and self-efficacy in the COVID-19 era. The authors should relate to such up-to-date studies (e.g.

Shacham, M., Hamama-Raz, Y., Kolerman, R., Mijiritsky, O., Ben-Ezra, M., & Mijiritsky, E. (2020). COVID-19 factors and psychological factors associated with elevated psychological distress among dentists and dental hygienists in Israel. International Journal of Environmental Research and Public Health, 17(8), 2900.

Consolo, U., Bellini, P., Bencivenni, D., Iani, C., & Checchi, V. (2020). Epidemiological Aspects and Psychological Reactions to COVID-19 of Dental Practitioners in the Northern Italy Districts of Modena and Reggio Emilia. International Journal of Environmental Research and Public Health, 17(10), 3459.)

Response: Thank you for your valuable comment. We revised the manuscript in the discussion section and included the relevant papers published up to date. We had not included them in our previous manuscript version since these interesting papers were not available and had not been published. 

7. "a text message invitation to take a survey was sent". How was it sent? WhatsApp? Were participants contacted more than once?

Response: Thank you for your valuable comment. We have added this sentence in the Sample and Setting section: "This invitation message was sent via Short Message Service (SMS) on national registered phone users and WhatsApp application to a list of all registered dentists in the general dental council. For those who did not complete the questionnaire within 10 days, the invitation message was resent again on March 15, 2020." Page7, Line179. 

8. Some parts of the questionnaire has very low number of items, e.g. perceived sustainability, along with internal reliability values rated as good. These should be added to the limitations of the study.

Response: Thank you for your valuable comment. We have included the low number of items for some constructs as a limitation in the discussion. However, the number of items in each construct was obtained according to the reliability and validity procedures described in the survey instruments section. At first, the number of items in each construct was high, but some items were removed during the validity and reliability procedure according to the expert panel’s opinions and to increase the value of Cronbach's alpha. Page17, Line 388.

9. In what language was the survey conducted? English, Arabic, Farsi, other?

Response: Thank you for your valuable comment. We have added this sentence to the design section: "This survey was conducted in Farsi." Page6, Line156.

10. Were the questions adopted from previous validated questionnaires?

Response: Thank you for your valuable comment. We have added this sentence in the survey instrument section: "Validated questionnaires used in previous studies were used to design and construct the EPPM scale." Page8, Line190.

11. "Significance level in all tests was considered at p<0.05". Perhaps the authors meant consideration at p≤0.05? Since in the results several parameters were

Response: Thank you for your kind comment. The sentence has been revised as: "Significance level in all tests was considered at p≤0.05." Page 10, Line 236.

12. "The proportion of high efficacy was significantly higher is employees with more than 5-years of previous work experience (P=0.04) and those older than 30 years of age (P=0.08)." "How come P=0.08 and P=0.05 are considered significant, if values were considered significant <0.05?

Response: Thank you for your valuable comment. We have deleted these sentences "and those older than 30 years of age (P=0.08)".

Response to Reviewer 3:

Abstract

1. English remains a concern. Please revise for language shortcomings, to facilitate reading. See, for example, "procedres".

 and high efficacy towards coronaviruse disease, respectively." Meaning is unclear, please revise. - "

The proportion of high efficacy was significantly higher (...)." Meaning remains unclear.

- "Data identified that those oral health care providers that were on the risk control response (...)." Again, what is risk control response?

- "(...) adopted preventive behaviors more strictly r than those on fear control response." Check sentence, and clarify the term "fear control response".

Response: Thank you for your comments. We have revised and improved the English language.

Intro

2. Meaning ov "observaning"?

Response: We have revised the sentence to not confuse the readers with the verb observing: By not following the hygienic recommendations, healthcare professionals will become unwanted carriers and transmit the disease to other patients, their family members and to the community. Page3, Line77.

3. Again, please revise for a clear language. Provide complete sentences. Note that the current draft must be thoroughly revised, and help of a native speaker is strongly encouraged. With the numerous (co-authors) of the current submission, quality of the submitted manuscript clearly would seem questionable. Remember that ALL authors must have read (and should have revised) this submitted draft.

 Response: We have revised the manuscript thoroughly for the English language.

4. "In infectious disease epidemic campaigns, main focus is on motivating fear." Meaning would seem unclear. Who motivates fear? Provide references.

Response: Thank you for your valuable comment. The sentence has been revised as: "Research has shown that risk messages distributed through health care providers can cause fear in patients and lead to positive changes in their attitude, intention and protective behavior responses." We also have provided reference. Page 4, Line 95-97.

5. Authors have missed to elaborated both aims and objectives. This section is not considered satisfying.

Response: Thank you for your valuable comment. We have added these sentences in the last paragraph of the introduction section:

“Due to the outbreak of coronavirus infection in Iran and the high importance of infection control in dentistry, our objectives were to answer the following questions:

1. What is the percentage of oral health care providers who are on fear control and danger control response?

2. What is the influence of different demographic characteristics on health behaviors among oral health care providers regarding COVID-19 epidemic?

Page5, Line 132.

6. A clear and indisputable null hypothesis would seem missing.

Response: Thank you for your valuable comment. We have added these sentences in the last paragraph of the introduction section:

“In addition, we addressed the following null hypotheses in our study:

- There are no associations between oral health care providers’ demographic characteristics and perceived threat and efficacy to COVID-19 epidemic.

-. There is no association between oral health behavior and perceived threat and efficacy to COVID-19 epidemic among oral health care providers.

- There are no associations between oral health care providers’ demographic characteristics and danger control responses to COVID-19 epidemic.” Page5, Lines 142-148.

Meths

7. Numerous aspects do not belong to the Methodology section here. Remember not to provide a literature review with this section. Please revise thoroughly, and provide references. Your explanation of EPPM should be transferred to the Intro section.

Response: Thank you for your valuable comment. We have transferred EPPM explanation to the Intro section. 

8. "In total, 300 (34.8%) completed electronic questionnaires were received." This would seem to present a result, right?

Response: Thank you for your valuable comment. We have transferred this sentence "In total, 300 (34.8%) completed electronic questionnaires were received." to the result section. Page 10, Line 242.

9. Please compare: "0.78 for "perceived severity" 0.78, 0.84 for "self-efficacy", (...)", and you will understand that it seem doubtful whether all co-authors have read and approved this manuscript.

Response: Thank you for your valuable comment. We have revised this section and explained in the methods section how exactly the questionnaire was designed:

The reliability of the questionnaire was assessed by calculating the internal reliability. Alpha Cronbach was 0.70 for "perceived sustainability", 0.78 for "perceived severity", 0.84 for "self-efficacy", 0.87 for "response efficacy" and 0.92 for "preventive behaviors. Page 8, Lines 190-196.

10. "was consisted of 30 questions which covered five constrcuts": Again, please revise thoroughly, and eliminate all language shortcomings.

Response: Thank you for your valuable comment. We have corrected the typo and revised the section thoroughly in the methods section. 

“We designed and constructed the EPPM scale based on questionnaires used by previous peer-reviewed studies (1,2). In these studies, the validity and reliability of the questionnaires were measured through Exploratory Factor Analysis (EFA), Confirmatory Factor Analysis (CFA), Cronbach’s alpha coefficient and the Intraclass Correlation Coefficient (ICC).

The validity measures and Cronbach’s alpha coefficient of the designed questionnaire in this study are similar to previous studies. However, the Cronbach's alpha constructs in this study were eventually re-examined and improved.” Page 8, Line 190-196.

1. Barati M, Bashirian S, Jenabi E, Khazaei S, Karimi-Shahanjarini A, Zareian S, et al. Factors Associated with Preventive Behaviours of COVID-19 among Hospital Staff in Iran in 2020: An Application of the Protection Motivation Theory. Journal of Hospital Infection. 2020.

2. Jahangiry L, Sarbakh P, Reihani P, Samei S, Sohrabi Z, Tavousi M, et al. Developing and validating the risk perceptions questionnaire for COVID-19 (Risk Precept COVID-19): an application of the extended parallel process model. 2020.

11. What is "susestbility"?

Response: Thank you for your valuable comment. It was a typo. We have corrected this word to susceptibility. 

Results

12. Again, this section has not been convincingly elaborated. Numerous typos should be eliminated, to facilitate reading.

Response: Thank you for your valuable comment. We have deleted duplicate results in result section and corrected the typos.

Disc

13. As with the other sections, proper elaborations would seem mandatory. See "efficacy" vs "efficiency".

Response: Thank you for your valuable comment. We have improved English language. 

14. "COVID 19 infection had been selected since it is currently an international public health problem." This is not considered"strength" of the submitted study/paper.

Response: Thank you for your valuable comment. We have deleted this sentence" First, COVID 19 infection had been selected since it is currently an international public health problem."

15. "Based on the results, specific target groups can be identified and the necessary interventions can be tailored according to the control processes." Please provide these information’s. What "specific target groups" and "what necessary interventions to be tailored" do you refer to?

Response: Thank you for your valuable comment. We have added this paragraph in this discussion section: 

“According to the results of this study, younger participants, those with less previous work experience, females, and dental hygienists were more motivated to control their fears than other groups.

Educational interventions on COVID-19 infection control guidelines that focus on increasing self-efficacy, such as self-efficacy model, self-regulation model can be designed to increase their perceived efficacy. Environmental interventions such as providing personal protective equipment (PPE) for dental students and Dental hygienists, and implementation of Safety and Hygiene Scheme 2020 (SHS 2020) in dental schools through PRECEDE-PROCEED Model are other possible suggestions.” Page 16, Lines 357-370.

16. What is "bias induced by social utility"? Please explain?

Response: Thank you for your valuable comment. Due to the self-reporting nature of the questionnaire, participants may not report the actual behavior but the behavior that would be desired by the society. 

The sentence has been revised as: "One of the limitations of this study was that questionnaires were self-administered and could induce social desirability. Since participants may not report the actual behavior but the behavior that would be desired by the society. “Page 16, Lines 375-378.

Concl

17. Please do not simply repeat your results here. Instead, provide a reasonable extension of your outcome.

Response: Thank you for your valuable comment. We have deleted this paragraph from the conclusion: "Women's danger control was lower than that of men. Dental students and dental hygienists were less motivated to control danger. Given that the coronavirus pandemic is an important public health issue, and the protection of medical staff such as dentists and patient safety in clinical care should be taken seriously, the placement of oral health care provider in the motivation of fear control, will lead to neglect of key health messages and actions or relevant behaviors to prevent COVID-19. Therefore, it is necessary to develop educational programs to increase the efficacy of oral health professionals in helping to prevent the spread of the disease." and these paragraphs have been replaced as follows:

"The results of this study showed how degrees of perceived threat and perceived efficacy affect oral health care providers’ willingness to perform recommended health behaviors. The findings showed that when perceived efficacy of the recommended health behaviors overcame the perceived threat, the likelihood of preventive health behaviors regarding COVID-19 increased. Therefore, a theory-based behavioral modification program can be developed based on gender among dental students and dental hygienists. Older oral health care providers and those with more years of experience require intense educational interventions to modify their hygienic behaviors compared to younger providers." Page 17, Lines 386-393.

18. Stick exclusively to your aims, and focus on your outcome. Do not provide general phrases or common places here.

Response: Thank you for your valuable comment. We have deleted this paragraph:" Women's danger control was lower than that of men. Dental students and dental hygienists were less motivated to control danger. Given that the coronavirus pandemic is an important public health issue, and the protection of medical staff such as dentists and patient safety in clinical care should be taken seriously, the placement of oral health care provider in the motivation of fear control, will lead to neglect of key health messages and actions or relevant behaviors to prevent COVID-19. Therefore, it is necessary to develop educational programs to increase the efficacy of oral health professionals in helping to prevent the spread of the disease." and these paragraphs have been replaced as follows:

"The results of this study showed how degrees of perceived threat and perceived efficacy affect oral health care providers’ willingness to perform recommended health behaviors. The findings showed that when perceived efficacy of the recommended health behaviors overcame the perceived threat, the likelihood of preventive health behaviors regarding COVID-19 increased. Therefore, a theory-based behavioral modification program can be developed based on gender among dental students and dental hygienists. Older oral health care providers and those with more years of experience require intense educational interventions to modify their hygienic behaviors compared to younger providers." Page 17, Lines 386-393.

Tables & Figs

19. Do not repeat or double your results.

Response: Thank you for your valuable comment. We have deleted duplicate results in result section.

---

## [Decision Letter · Decision Letter 1]

20 Jul 2020

PONE-D-20-14093R1

Fear Control and Danger Control Amid COVID-19 Dental Crisis: Application of the Extended Parallel Process Model

PLOS ONE

Dear Dr. Bashirian,

Thank you for submitting your manuscript to PLOS ONE. After careful consideration, we feel that your submitted paper might have merit but still does not meet PLOS ONE’s publication criteria as it currently stands.

Having intensively reviewed your draft, our external referees have indicated that most revisions would be satisfying; however, I still have found major shortcomings, please see R #3 comments. All in all, the indicated shortcomings are considered reasonable with regard to both PLOS ONE's quality standards and our readership's expectations. Therefore, we invite you to submit a revised version of the manuscript that addresses each and every point raised during the review process. Please pay special attention to language, structure and coherence of your full text, and to uniform formatting of your References section (please stick to our Authors' Guidelines).

We look forward to receiving your revised manuscript.

Kind regards,

Andrej M Kielbassa

Academic Editor

PLOS ONE

Reviewers' comments:

Reviewer's Responses to Questions

**Comments to the Author**

1. If the authors have adequately addressed your comments raised in a previous round of review and you feel that this manuscript is now acceptable for publication, you may indicate that here to bypass the “Comments to the Author” section, enter your conflict of interest statement in the “Confidential to Editor” section, and submit your "Accept" recommendation.

Reviewer #1: All comments have been addressed

Reviewer #2: All comments have been addressed

Reviewer #3: All comments have been addressed

2. Is the manuscript technically sound, and do the data support the conclusions?

Reviewer #1: Yes

Reviewer #2: Yes

Reviewer #3: No

3. Has the statistical analysis been performed appropriately and rigorously? 

Reviewer #1: Yes

Reviewer #2: Yes

Reviewer #3: Yes

4. Have the authors made all data underlying the findings in their manuscript fully available?

Reviewer #1: Yes

Reviewer #2: Yes

Reviewer #3: No

5. Is the manuscript presented in an intelligible fashion and written in standard English?

Reviewer #1: Yes

Reviewer #2: Yes

Reviewer #3: No

6. Review Comments to the Author

Reviewer #1: Dear Authors

I have quickly reviewed and found that all authors done very well. All the reviewer suggestions look addressed nicely.

If there is any grammatical error correct carefully in the gallery proof from the journal.

Reviewer #2: Thank you for improving your manuscript and adressing all of my comments. This topic is utterly important and should be published in order to provide healthcare policy makers with sufficient information regarding it.

Reviewer #3: This revised and re-submitted draft has been considerably improved, no doubt. However, several aspects still would not seem satisfying, and language/writing has not been convincingly revised. Please note that this is the authors' responsibility, and there are many authors with this manuscript who should take care.

- Please note that one/two sentences do not constitute one paragraph. Please revise carefully, and summarize your thoughts in reasonable sections. Revise throughout your text.

- "As of 28th of March, COVID19 has been diagnosed in 199 countries with 614,231 laboratory-confirmed cases and 28,240 deaths (2) ." Please update your information. Meanwhile, it is July.

- Revise thoroughly for spacebar use.

- "(...)COVID-19 epidemic.” " Please revise for quotation marks not considered necessary.

- p≤0.05 must read p<0.05.

- All p values must be given on a three-digit basis. Revise thoroughly.

- Again, please revise carefully. "efficacy(LT/HE)" should read "efficacy (LT/HE)".

- Same with "Table 3. Health (...)".

- Same with "P.Value". Delete your dot.

- Please compare: behavior2:, behavior 4:, "(gowns , apron, (...)".

- Please revise: "disease -related guidelines", "self-regulation model , etc.," What is "etc"? Why closing this sentence with a comma?

- Referees still not uniformly formatted. Please see Authors' Guidelines.

7. PLOS authors have the option to publish the peer review history of their article (what does this mean?). If published, this will include your full peer review and any attached files.

Reviewer #1: No

Reviewer #2: **Yes: **Dr. Maayan Shacham

Reviewer #3: No

---

## [Author Response · Author response to Decision Letter 1]

25 Jul 2020

Title: Fear Control and Danger Control Amid COVID-19 Dental Crisis: Application of the Extended Parallel Process Model

Authors:

Samaneh Shirahmadi (shirahmadi_s@yahoo.com)

Shabnam Seyedzadeh-Sabounchi, 

Salman Khazaei, 

Saeid Bashirian, 

Amir Farhang Miresmæili, 

Zeinab Bayat, 

Behzad Houshmand

Hasan Semyari

Majid Barati, 

Ensiyeh Jenabi, 

Fakhreddin Heidarian, 

Sepideh Zareian, 

Mohammad Kheirandish, 

Neda Dadae

Version: 2 Date: 25 July 2020

Author's response to reviews: see over

We thank all the Reviewers for their valuable feedback and taking the time to provide useful comments to improve our manuscript entitled “Fear Control and Danger Control amid COVID-19 Dental Crisis: Application of the Extended Parallel Process Model”. 

Based on the constructive comments the following changes have been made. 

Response to Reviewer 3:

1. Please note that one/two sentences do not constitute one paragraph. Please revise carefully, and summarize your thoughts in reasonable sections. Revise throughout your text.

Response: Thank you for your comments. We have incorporated this advice in all sections of the manuscript and revised accordingly.

2. "As of 28th of March, COVID19 has been diagnosed in 199 countries with 614,231 laboratory-confirmed cases and 28,240 deaths (2)." Please update your information. Meanwhile, it is July.

Response: Thank you for your valuable comment. The sentence has been revised as: "As of 20th of July, COVID19 has been diagnosed in 213 countries with 14,855,107 laboratory-confirmed cases and 613,248 deaths". Page 3, Line 70.

And "revised these sentences "As of 25th of March 2020, COVID 19 had been diagnosed in Iran with 27017 laboratory-confirmed cases and 2077 deaths had been recorded": As of 21st of July 2020, COVID 19 had been diagnosed in Iran with 276,202 laboratory-confirmed cases and 14,405 deaths had been recorded." Page 13, Line 300.

3. Revise thoroughly for spacebar use.

Response: Thank you for your comments. We have revised the manuscript thoroughly for the proper use of spacebar.

4. "(...)COVID-19 epidemic.” "Please revise for quotation marks not considered necessary.

Response: Thank you for your comments. We have revised and removed unnecessary quotation marks.

5. p≤0.05 must read p<0.05.

Response: Significance level in all tests was considered at p<0.05." Page 8, Line 202.

6. All p values must be given on a three-digit basis. Revise thoroughly.

Response: Thank you for your comments. We have revised all p values.

7. Again, please revise carefully. "efficacy(LT/HE)" should read "efficacy (LT/HE)".

Response: Thank you for your comments. We have revised our manuscript.

8. Same with "Table 3. Health (...)".

Response: Thank you for your comments. We have revised our manuscript.

9. Same with "P.Value". Delete your dot.

Response: Thank you for your comments. We have revised all p values.

10. Please compare: behavior2:, behavior 4:, "(gowns , apron, (...)".

Response: Thank you for your valuable comment. These two behaviors reflect two different behaviors of protecting the eyes and the other is about wearing gloves and protective clothing which we have revised in the manuscript to clearly state these two different behaviors for the readers " use protection for eyes such as goggles, masks or shields during all treatments and use of protective uniforms (gown or protective clothing, protective gloves) during all treatment steps

11. Please revise: "disease -related guidelines", "self-regulation model, etc.," What is "etc"? Why closing this sentence with a comma?

Response: Thank you for your comments. The sentence has been revised as:" These interventions include educating providers on corona virus disease-related guidelines and recommendations for conducting health behaviors using health education models. The focus of the models can be directed towards increasing self-efficacy, such as self-efficacy model, self-regulation model and using implementation intention theory. Page 14, Line 323.

12. Referees still not uniformly formatted. Please see Authors' Guidelines.

Response: Thank you for your comments. We have revised all references.

---

## [Decision Letter · Decision Letter 2]

29 Jul 2020

Fear Control and Danger Control Amid COVID-19 Dental Crisis: Application of the Extended Parallel Process Model

PONE-D-20-14093R2

Dear Dr. Bashirian,

We’re pleased to inform you that your manuscript has been judged scientifically suitable for publication and will be formally accepted for publication once it meets all outstanding technical requirements.

Kind regards,

Andrej M Kielbassa, Prof. Dr. med. dent. Dr. h. c.

Academic Editor

PLOS ONE

Additional Editor Comments (optional):

Reviewers' comments:

Reviewer's Responses to Questions

**Comments to the Author**

1. If the authors have adequately addressed your comments raised in a previous round of review and you feel that this manuscript is now acceptable for publication, you may indicate that here to bypass the “Comments to the Author” section, enter your conflict of interest statement in the “Confidential to Editor” section, and submit your "Accept" recommendation.

Reviewer #3: All comments have been addressed

2. Is the manuscript technically sound, and do the data support the conclusions?

Reviewer #3: Yes

3. Has the statistical analysis been performed appropriately and rigorously? 

Reviewer #3: Yes

4. Have the authors made all data underlying the findings in their manuscript fully available?

Reviewer #3: Yes

5. Is the manuscript presented in an intelligible fashion and written in standard English?

Reviewer #3: Yes

6. Review Comments to the Author

Reviewer #3: This resubmitted draft would seem ready to proceed. All shortcomings having been identified recently have been addressed.

7. PLOS authors have the option to publish the peer review history of their article (what does this mean?). If published, this will include your full peer review and any attached files.

Reviewer #3: No

---

## [Editor Report · Acceptance letter]

3 Aug 2020

PONE-D-20-14093R2 

Fear Control and Danger Control Amid COVID-19 Dental Crisis: Application of the Extended Parallel Process Model 

Dear Dr. Bashirian:

I'm pleased to inform you that your manuscript has been deemed suitable for publication in PLOS ONE. Congratulations! Your manuscript is now with our production department. 

Kind regards, 

on behalf of

Prof. Dr. med. dent. Dr. h. c. Andrej M Kielbassa 

Academic Editor

PLOS ONE